# Global epidemiological and genetic characteristics of carbapenem-resistant *Escherichia coli* carrying *bla*$_{IMP}$

Yueyue Hu,[1,2] Chengjin Wu,[2] Jiayao Yao,[2] Jingyi Guo,[2] Jie Sheng,[2] Xinru Wang,[2] Longjie Zhou,[2] Xinyan Tang,[2] Haotian Xu,[2] Yunsong Yu,[1,2] Xi Li,[1,2] Xinhong Han[3]

**ABSTRACT** Carbapenem-resistant *Escherichia coli* (CREC) is a serious global health threat. While *bla*$_{NDM}$-positive and *bla*$_{KPC}$-positive CRECs are well-characterized, the genomic and epidemiological features of *bla*$_{IMP}$-positive CRECs remain poorly understood. This study provides a comprehensive analysis of *bla*$_{IMP}$-positive CRECs by integrating local and global surveillance data. We characterized nine *bla*$_{IMP-4}$-positive CRECs from 94 CRECs collected from four tertiary hospitals in Zhejiang, China (2017–2024), using whole-genome sequencing and assays for antimicrobial susceptibility, plasmid transfer, stability, and fitness. The nine isolates were multidrug-resistant but susceptible to colistin, tigecycline, and amikacin. Seven diverse sequence types (STs) were identified, with high-risk ST1193 clones being first detected. The isolates had similar *bla*$_{IMP}$-carrying IncN plasmids with high conjugation frequencies ($5.3 \times 10^{-2}$ to $8.7 \times 10^{-1}$), 100% stability after 100 generations, and minimal fitness cost, facilitating regional dissemination. In addition to plasmids, other mobile genetic elements, like class 1 integrons, transposons, and insertion sequences, can also facilitate the horizontal transfer of *bla*$_{IMP}$ genes. Global analysis of 340 *bla*$_{IMP}$-positive CRECs revealed high genetic diversity (82 STs), with ST131 predominant (36.18%). Clonal spread was observed for ST131 and ST216. Significant positive correlation existed between antibiotic resistance genes and plasmid replicons ($r = 0.5205$, $P < 0.001$), highlighting plasmids' role in resistance transmission. China, Japan, and Australia were the top 3 contributing countries, which also showed relatively diverse resistance genes but similar virulence factors. These findings emphasize the evolving risk of *bla*$_{IMP}$-positive CRECs and underscore the need for sustained genomic surveillance.

**IMPORTANCE** Compared with *bla*$_{NDM}$-positive and *bla*$_{KPC}$-positive carbapenem-resistant *Escherichia coli* (CREC), *bla*$_{IMP}$-positive CRECs are often overlooked. This study emphasized IncN plasmids, which, with their strong transferability, stability, and low fitness cost, can serve as vectors for the widespread dissemination of *bla*$_{IMP}$ genes. Additionally, the study elucidated the global epidemiological status, transmission features, and potential threats of *bla*$_{IMP}$-positive CRECs through a large-scale analysis of publicly available genomic data. These findings not only facilitate a better understanding of the genetic dynamics and transmission pathways of *bla*$_{IMP}$ but also help develop effective regulatory measures and antibiotic stewardship programs to alleviate the increasingly severe burden of antimicrobial resistance.

**KEYWORDS** *bla*$_{IMP}$, IncN plasmid, *Escherichia coli*, multidrug-resistant, genomic epidemiology

**Peer Reviewer** Fuping Hu, Institute of Antibiotics, Huashan Hospital, Fudan University, Shanghai, China

Address correspondence to Xinhong Han, hanxh@zju.edu.cn, Xi Li, lixi_0611@163.com, or Yunsong Yu, yvys119@zju.edu.cn.

Yueyue Hu and Chengjin Wu contributed equally to this article. The co-first authors order was determined based on mutual agreement upon by the authors.

The authors declare no conflict of interest.

Carbapenems, with their good permeability and strong antimicrobial activity, were often used to treat serious infections with multidrug-resistant Gram-negative organisms (1). However, in recent years, the worldwide overuse of β-lactam

drugs (especially carbapenems) has led to increasing reports of carbapenem-resistant *Enterobacteriaceae* and even the rapid spread of nosocomial infections (2). The China Antimicrobial Surveillance Network 2024 (https://www.chinets.com/) report showed that *Escherichia coli* was the primary clinically isolated strain. Carbapenem-resistant *Escherichia coli* (CREC) slightly increased to 3.6%, up by 0.5% from last year. So the total number of CRECs should not be overlooked. CRECs can not only persist in hospital environments and spread among patients and healthcare workers, but also disseminate widely in sewage, surface water, vegetation soil, as well as in animals, food, and healthy populations, posing a significant threat to public health (3–6).

The main mechanism of carbapenem resistance in *E. coli* is the production of carbapenemases, among which IMP-type metallo-β-lactamases are one of the prevalent acquired carbapenemases. The IMP-1 enzyme was detected in *Pseudomonas aeruginosa* by Japanese scholars in 1991 (7). As time progressed, more and more IMP variants appeared, including IMP-2 (8), IMP-4 (9), IMP-6 (10), and so on. Currently, there are 111 IMP variants (http://bldb.eu/, last updated on 18 September 2025). IMP variants exhibit different hydrolysis efficiencies, and their diversity enables bacteria to continuously adapt to complex environments, posing severe challenges to antibiotic stewardship (11–13).

Some $bla_{IMP}$-positive CRECs have low levels of imipenem resistance, which leads to an underestimation of IMP prevalence; for instance, IMP-6 remains susceptible to imipenem (14). The $bla_{IMP}$ gene in *E. coli* usually co-exists with $bla_{CTX-M}$ resistance genes or other resistance mechanisms, leading to resistance to multiple antibiotics such as cephalosporins, aminoglycosides, quinolones, carbapenems, and novel β-lactamase inhibitor combinations (e.g., ceftazidime-avibactam). In addition, most $bla_{IMP}$ genes are located on plasmids and frequently associated with integrons, insertion sequences, and transposons, which mediate their efficient horizontal transfer within the *Enterobacteriaceae* (11). Previous studies showed that $bla_{IMP}$ genes not only had a higher prevalence in Asian countries and Australia, but also had the ability to spread worldwide (15, 16). The $bla_{IMP}$-positive CRECs, with high transmissibility and drug resistance, pose severe challenges to clinical treatment and public health.

However, a detailed characterization of $bla_{IMP}$-positive CRECs and a systematic global epidemiological analysis remain limited. This study aims to address this gap by comprehensively characterizing the nine $bla_{IMP-4}$-positive CREC strains isolated from Zhejiang, China, with a particular focus on their plasmid transmission properties, and by elucidating the global epidemiological status, transmission features, and potential threats of $bla_{IMP}$-positive CRECs through a large-scale analysis of publicly available genomic data. This study will enable us to understand their distribution and transmission and help us formulate effective regulatory measures.

## RESULTS

### Characteristics of the nine CRECs

Among the 94 CRECs, 77 isolates were NDM carbapenemase producers, nine isolates carried the $bla_{IMP-4}$ gene, and four isolates carried the $bla_{KPC-2}$ gene. In addition, among the 94 CRECs, 4 isolates also exhibited carbapenem resistance mediated by other uncharacterized mechanisms. In this study, the nine $bla_{IMP-4}$-positive CRECs were further studied. The patients were all over 60 years old, and the strains were mainly isolated from bile, urine, and rectal swabs (Table S1). After whole-genome sequencing (WGS), the nine strains all carried IncN plasmids, $bla_{IMP-4}$ and *qnrS1* resistance genes. The majority of these isolates also contained extended-spectrum β-lactamase gene ($bla_{CTX-M}$), aminoglycoside genes (*aph(6)-Id*, *aph(3")-Ib*), sulfonamide gene (*sul*), trimethoprim gene (*dfrA*), and tetracycline gene (*tet(A)*). In addition, all isolates carried multiple virulence genes, including those for bacterial biofilm formation, adhesion, and iron uptake; genes encoding enterotoxin and an effector protein (which interferes with signaling pathways); and genes associated with bacterial immune evasion mechanisms (Fig. 1; Table S2). A phylogenetic tree was constructed based on genomic single nucleotide polymorphisms

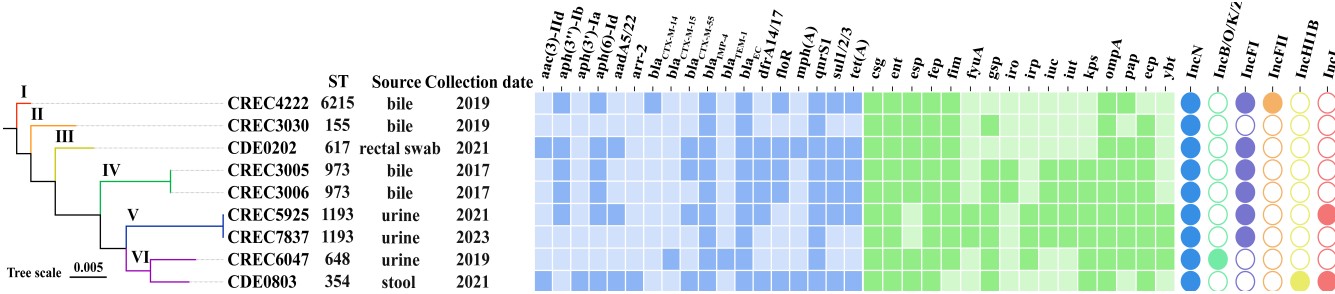

**FIG 1** Characteristics of the nine CRECs. Distribution of representative ARGs, VFs, plasmid Inc. types (not all are shown here), and phylogenetic relationships of the nine *Escherichia coli* from this study. The solid figures represent the presence of the corresponding ARGs, VFs, and plasmid replicons.

(SNPs), and the nine strains could be divided into six distinct phylogenetic clades (Fig. 1). According to previous studies, clonal dissemination can be defined based on a pairwise SNP distance of ≤200 between *E. coli* isolates (17, 18). Combined with multi-locus sequence typing (MLST), it was found that CREC5925 and CREC7387 were the same clone (ST1193), and CREC3005 and CREC3006 shared the same clone (ST973). For the remaining strains, their sequence types (STs) were relatively scattered, with a large difference in SNPs (Fig. S1).

## Antimicrobial susceptibility testing

According to minimal inhibitory concentrations (MICs), the results of antimicrobial susceptibility testing showed that all collected strains were resistant to ceftazidime, cefepime, ceftazidime-avibactam, and ertapenem. Except for CREC3030, the remaining strains (88.89%) were resistant to ciprofloxacin. For meropenem, only CREC5925 showed intermediate susceptibility, while the other strains had low-level resistance (MICs: 4 mg/L–16 mg/L). Additionally, three strains (CREC5925, CREC3030, and CDE0803) exhibited intermediate susceptibility to imipenem, and the rest had low-level resistance. All strains were sensitive to polymyxin, tigecycline, and amikacin, providing potential options for treatment (Table 1). Except for CREC7837, which failed to conjugate successfully, the

**TABLE 1** Antimicrobial susceptibilities of nine *Escherichia coli* strains and eight transconjugants conducted by *Escherichia coli* and *E. coli* J53[a]

| Isolates | MICs (µg/mL) | | | | | | | | | |
|---|---|---|---|---|---|---|---|---|---|---|
| | ETP | FEP | MEM | IPM | CAZ | CZA | AK | CST | TGC | CIP |
| CREC4222 | 64 | >128 | 16 | 8 | >128 | >128 | 2 | <0.125 | 0.0625 | 32 |
| CREC6047 | 64 | >128 | 16 | 4 | >128 | 128 | 1 | <0.125 | 0.5 | 32 |
| CREC5925 | 8 | >128 | 2 | 2 | >128 | >128 | 1 | <0.125 | 0.0625 | 32 |
| CREC3030 | 32 | 32 | 16 | 2 | >128 | >128 | 4 | <0.125 | 0.5 | 0.25 |
| CREC3005 | 32 | 64 | 16 | 4 | >128 | >128 | 2 | <0.125 | 0.25 | 32 |
| CREC3006 | 32 | 16 | 8 | 4 | >128 | >128 | 1 | <0.125 | 0.5 | 32 |
| CDE0202 | 128 | >128 | 32 | 8 | >128 | >128 | 1 | <0.125 | 0.5 | 32 |
| CREC7837 | 32 | 128 | 8 | 4 | >128 | >128 | 2 | <0.125 | 0.0625 | 32 |
| J53/pIMP-CREC4222 | 16 | 64 | 16 | 4 | >128 | >128 | 1 | <0.125 | 0.125 | 0.25 |
| J53/pIMP-CREC6047 | 16 | 128 | 16 | 4 | >128 | >128 | 1 | <0.125 | 0.125 | 0.5 |
| J53/pIMP-CREC5925 | 8 | >128 | 2 | 2 | >128 | 64 | 4 | <0.125 | 0.125 | 0.25 |
| J53/pIMP-CREC3030 | 8 | 4 | 2 | 1 | >128 | >128 | 2 | <0.125 | 0.5 | 0.25 |
| J53/pIMP-CREC3005 | 32 | 128 | 16 | 4 | >128 | >128 | 2 | <0.125 | 0.5 | 0.25 |
| J53/pIMP-CREC3006 | 32 | 16 | 8 | 2 | >128 | >128 | 0.5 | <0.125 | 0.5 | 0.5 |
| J53/pIMP-CDEC0202 | 16 | 16 | 8 | 2 | >128 | >128 | 0.5 | <0.125 | 0.25 | 0.25 |
| J53/pIMP-CDEC0803 | 8 | 4 | 8 | 2 | >128 | >128 | 1 | <0.125 | 0.25 | <0.125 |
| J53 | <0.125 | <0.125 | 0.25 | 0.25 | 0.5 | 0.5 | 1 | <0.125 | 0.5 | <0.125 |
| ATCC25922 | <0.125 | <0.125 | <0.125 | <0.125 | 0.25 | 0.25 | 1 | <0.125 | <0.125 | <0.125 |

[a]ETP, ertapenem; FEP, cefepime; MEM, neropenem; IPM, imipenem; CAZ, ceftazidime; CZA, ceftazidime-avibactam; AK, amikacin; CST, colistin; TGC, tigecycline; CIP, ciprofloxacin.

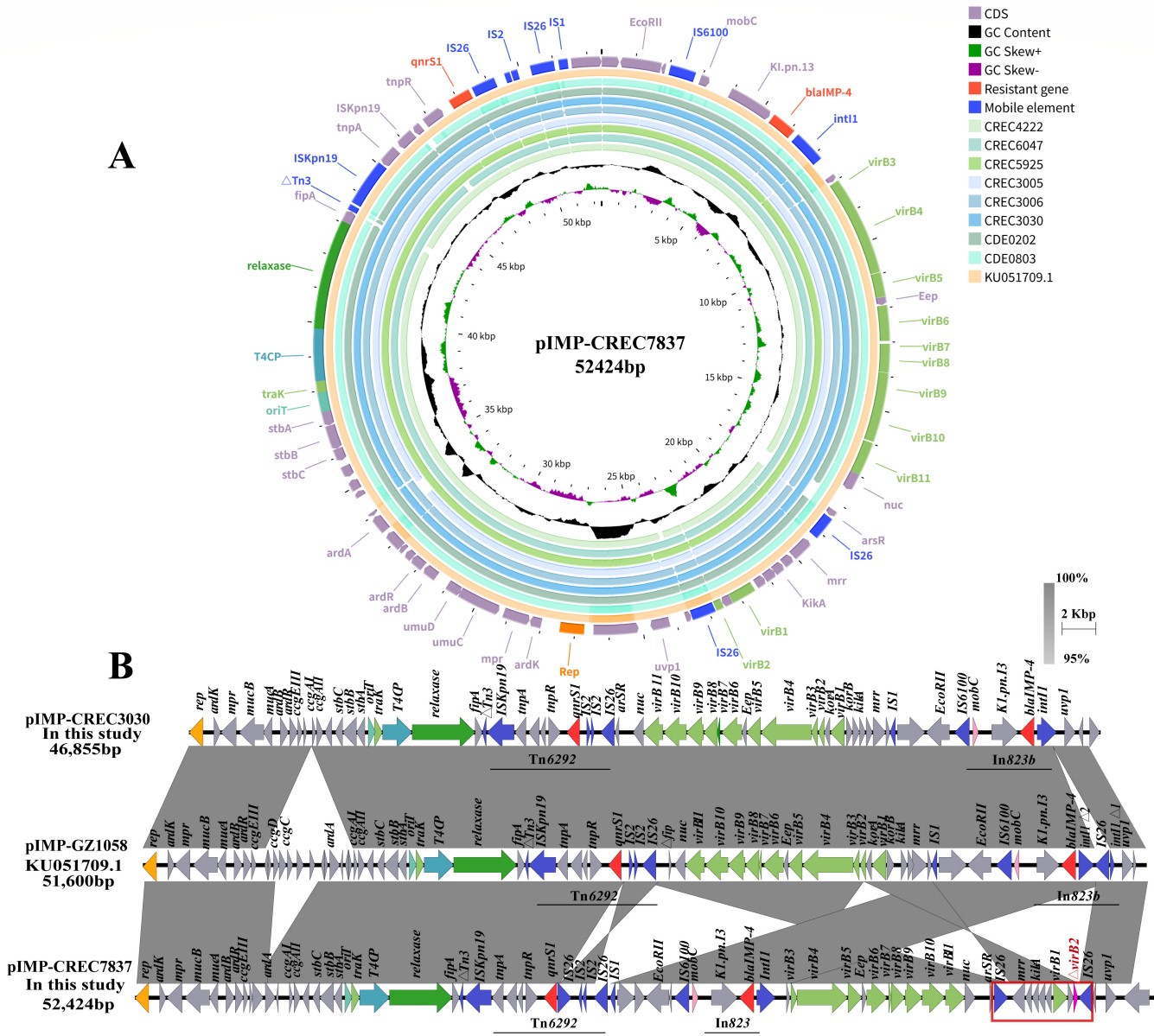

**FIG 2** Genomic characteristics of the *bla*IMP-4 gene in our strains. (A) Circular sequence alignment of plasmids bearing *bla*IMP-4. The pIMP-CREC7837 plasmid served as a reference for alignment with the other eight *bla*IMP-4 CRECs in this study and pIMP-GZ1058 (KU051709.1), similar plasmids available from the NCBI database. (B) Linear sequence alignment. Genetic environment of *bla*IMP-4 on pIMP-CREC7837, pIMP-GZ1058, and pIMP-CREC3030 plasmids. The regions with highly similar sequences were marked in gray (nucleotide identity of ≥95%). The *bla*IMP-4 and *qnrS1* genes are shown in red; mobile elements were drawn in blue.

*bla*IMP-4 genes of the other eight strains (88.89%, 8/9) were verified to have successfully conjugated to the recipient strain *E. coli* J53 (Table 1).

## Genomic characteristics of *bla*IMP-4-carrying IncN plasmids

In plasmid map structure comparison, nine strains contained *bla*IMP-4-carrying IncN plasmids with high consistency and similar backbone structures (Fig. 2A). We further performed third-generation sequencing on the CREC7837 strain. In NCBI BLAST, the available pIMP-CREC7837 and pIMP-CREC3030 plasmid sequences exhibited 99.99% nucleotide identity and 100% coverage with pIMP-GZ1058 (KU051709.1) from *E. coli* (Fig. 2B). These three IncN plasmids featured *rep*, *mobC*, *stbABC*, *ardABRK*, *mucAB,* and *kikA* genes essential for replication and maintenance. In addition, a conjugation transfer

system including origin of transfer site (*oriT*), relaxase, the type IV coupling proteins (T4CPs), and the type IV secretion system (T4SS) (*virB1–11*) was found. It is noteworthy that the *virB2* gene in the pIMP-CREC7837 plasmid was partially deleted. The IncN plasmid carried two resistant regions, including the $bla_{IMP-4}$ located on a class 1 integron and the *qnrS1* located on the Tn*6292* transposon. In pIMP-CREC7837 and pIMP-CREC3030 plasmids, the *intI1* integrase of class 1 integron (In*823*) captured the $bla_{IMP-4}$ resistance gene cassette, tandem with group II intron *K1.pn.I3*, a mobilization protein gene (*mobC*), and IS*6100* insertion sequences to form the structure of *intI1*-$bla_{IMP-4}$-*K1.pn.I3*-*mobC*-IS*6100*. The *qnrS1* was detected on Tn*6292*, which was a complex transposon with a core structure consisting of the following components: IS*Kpn19*-*tnpA*-*tnpR*-*qnrS1*-IS*2*-IS*26* (Fig. 2B).

These three plasmids also showed differences: (i) In the pIMP-CREC7837 plasmid, the insertion of IS*26* on both sides of *virB1* and Δ*virB2* led to the rearrangement of T4SS genes. And the IS*26* inserted downstream of the 209 bp sequence site of the *virB2* gene resulted in the deletion of the subsequent sequence, which may lead to the destruction of *virB2* gene integrity, loss of conjugative pilus function, and finally the failure of conjugation. (ii) In pIMP-CREC7837 and pIMP-CREC3030, the *intI1* integrase was intact, whereas in pIMP-GZ1058, *intI1* was truncated by IS*26* to form the In*823b* structure. (iii) In addition, due to IS*26* inserted between *qnrS1* and IS*2*, gene rearrangement occurred downstream of *qnrS1* in the pIMP-CREC7837 plasmid (Fig. 2B).

## Plasmid transfer capacity, plasmid stability, and fitness cost

Plasmid conjugation frequency assay showed the $bla_{IMP-4}$-carrying IncN plasmids were transferred by conjugation to the *E. coli* J53 recipient at high frequencies of $5.3 \times 10^{-2}$ to $8.7 \times 10^{-1}$ per donor cell. Due to the nine strains harbored similar $bla_{IMP-4}$-carrying IncN plasmids, we randomly selected one J53/pIMP-CREC3006 transconjugant to assess the fitness cost and plasmid stability. The results showed that there was no statistically significant difference in the relative growth rate between the J53/pIMP-CREC3006 transconjugant and *E. coli* J53 ($P > 0.05$). But a statistically significant difference was observed in the area under the growth curve (AUC) at 20 h (decreased by approximately 17.73%, $P < 0.05$). These results collectively indicated that the transconjugant carrying the IncN plasmid had a slight fitness cost (Fig. 3). In addition, the J53/pIMP-CREC3006 transconjugant was cultured for 10 consecutive days without antibiotics, and the IncN plasmid remained 100% after 100 generations. These results indicated that the plasmid exhibited high stability, strong transfer ability, and slight fitness cost, which facilitated the dissemination of the $bla_{IMP-4}$ genes.

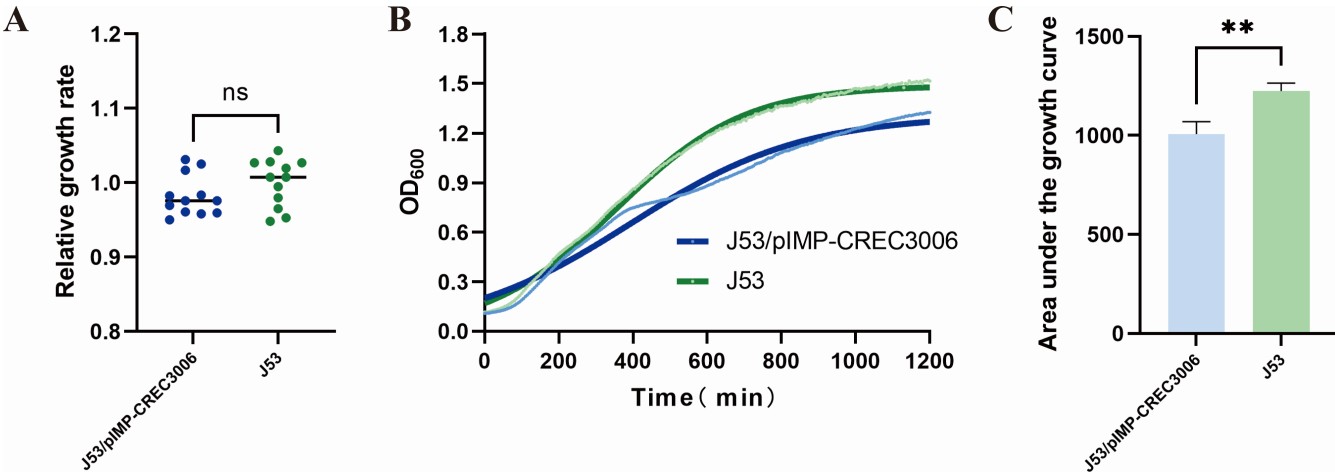

**FIG 3** Growth profile of the recipient strain *E. coli* J53 and transconjugant *E. coli* J53/pIMP-CREC3006 (A–C). (A) Relative growth rate. (B) Growth curve. (C) AUC. This experiment was performed in four replicates and repeated three times. (ns > 0.05; *$P < 0.05$; **$P < 0.01$; unpaired *t* tests with F test).

## Core genetic characterization of the *bla*$_{IMP}$ in *E. coli*

The *bla*$_{IMP}$ gene was usually located on the integron within a plasmid, leading to the horizontal transfer of resistance genes. Therefore, we further explored the *bla*$_{IMP}$ gene context in *E. coli*. All available complete genome sequences of *bla*$_{IMP}$-positive *E. coli* at NCBI were downloaded. The schematic plot contained five *bla*$_{IMP}$ genotypes: *bla*$_{IMP-1}$, *bla*$_{IMP-4}$, *bla*$_{IMP-6}$, *bla*$_{IMP-14}$, and *bla*$_{IMP-27}$ (Fig. 4A). It was worth mentioning that the *bla*$_{IMP-27}$ gene was located in the chromosomal class 2 integron with *intI2*-IS*Vsa5*-*bla*$_{IMP-27}$-*sat2*-*aadA1*-Tn*7*-like structure. Other *bla*$_{IMP}$ genes were mainly located in class 1 integrons of plasmids (including IncN, IncA/C2, IncHI2/IncN, etc). The classical backbone of class 1 integrons consisted of two conserved regions and the middle variable region. The 5′ conserved segment included the *intI1* gene. The 3′ conserved segment contained the *qacEΔ1* gene and *sul1* gene (19). The middle variable region was the most flexible part of the integron, harboring one or more gene cassettes. The specific classification of integrons was related to the composition of these gene cassettes. In*687*, In*798*, In*722*, In*809*, In*823*, and In*823b* were described in *E. coli*, with In*722* (IS*Kpn22*−) located on the IncN plasmid, and In*722 (ISKpn22+)* on the chromosome (Fig. 4A). We have further sorted out the following eight integron types containing the *bla*$_{IMP-4}$ gene, including In*586*, In*809*, In*823*, In*823b*, In*1377*, In*1456*, In*1460*, and In*1589* (Fig. 4B). In*809* was identified with the structure of *bla*$_{IMP-4}$-*qacG2*-*aacA4*-*catB3*-*qacEΔ1*-*sul1* in various plasmids. In*823* and In*823b* were usually discovered in the IncN plasmid. In these integrons, the *bla*$_{IMP}$ genes usually co-exist with one or more resistance genes, especially the aminoglycoside resistance gene *aacA4*. The majority of *qacEΔ1* gene and *sul1* gene are retained.

## Global surveillance of *bla*$_{IMP}$-positive CRECs

To further monitor the global *bla*$_{IMP}$-positive CRECs, we assessed all available 340 genomes of *bla*$_{IMP}$-positive CRECs in GenBank (Table S3). These isolates were mainly from 17 countries, with Japan accounting for the highest proportion of 40.29% (137/340), followed by Australia 31.18% (106/340), and China 9.71% (33/340). The USA, UK, and Singapore also had some distribution, and other countries were scattered (Fig. 5A). Among the CRECs, 13 *bla*$_{IMP}$ variants were identified, with *bla*$_{IMP-4}$ accounting for 39.41% (134/340), followed by *bla*$_{IMP-6}$ (32.65%, 111/340) and *bla*$_{IMP-1}$ (18.53%, 63/340). Specifically, *bla*$_{IMP-6}$ was the dominant variant in Japan, accounting for approximately 73.72% (101/137), followed by *bla*$_{IMP-1}$ (23.36%, 32/137). In Australia, the *bla*$_{IMP-4}$ variant was predominant, representing 95.28% (101/106). In China, *bla*$_{IMP-4}$ was also the main variant, accounting for 90.91% (30/33). We also found *bla*$_{IMP-27}$ and *bla*$_{IMP-64}$ CRECs with regional endemicity, which were isolated in the USA. From 2004 to 2010, few *bla*$_{IMP}$-positive *E. coli* strains were reported. Between 2011 and 2021, the overall detection rate remained relatively high with fluctuations, dominated by *bla*$_{IMP-1}$, *bla*$_{IMP-4}$, and *bla*$_{IMP-6}$ variants, while other variants emerged sequentially. Notably, the number of *bla*$_{IMP-4}$ in Australia surged in 2012, and the number of *bla*$_{IMP-6}$ in Japan surged in both 2013 and 2015 (Fig. 5B). Among the *bla*$_{IMP}$-positive CRECs, clinical sources accounted for 77.35% (263/340), predominantly *bla*$_{IMP-6}$ (39.16%, 103/263), *bla*$_{IMP-4}$ (32.32%, 85/263), and *bla*$_{IMP-1}$ (21.67%, 57/263). Animal sources accounted for 15.59% (53/340), with *bla*$_{IMP-4}$ accounting for 86.79% (46/53), followed by *bla*$_{IMP-27}$ and *bla*$_{IMP-38}$, which had a strong correlation with animal origin. Regarding environmental sources, one *E. coli* strain carrying both *bla*$_{IMP-1}$ and *bla*$_{NDM-5}$ genes was detected in a river in China in 2018, and another seven *E. coli* strains with *bla*$_{IMP-64}$ all originated from housing environments of animals in the USA (Fig. 5B). This showed that *E. coli* in animals and environments can also serve as a reservoir for *bla*$_{IMP}$ resistance genes.

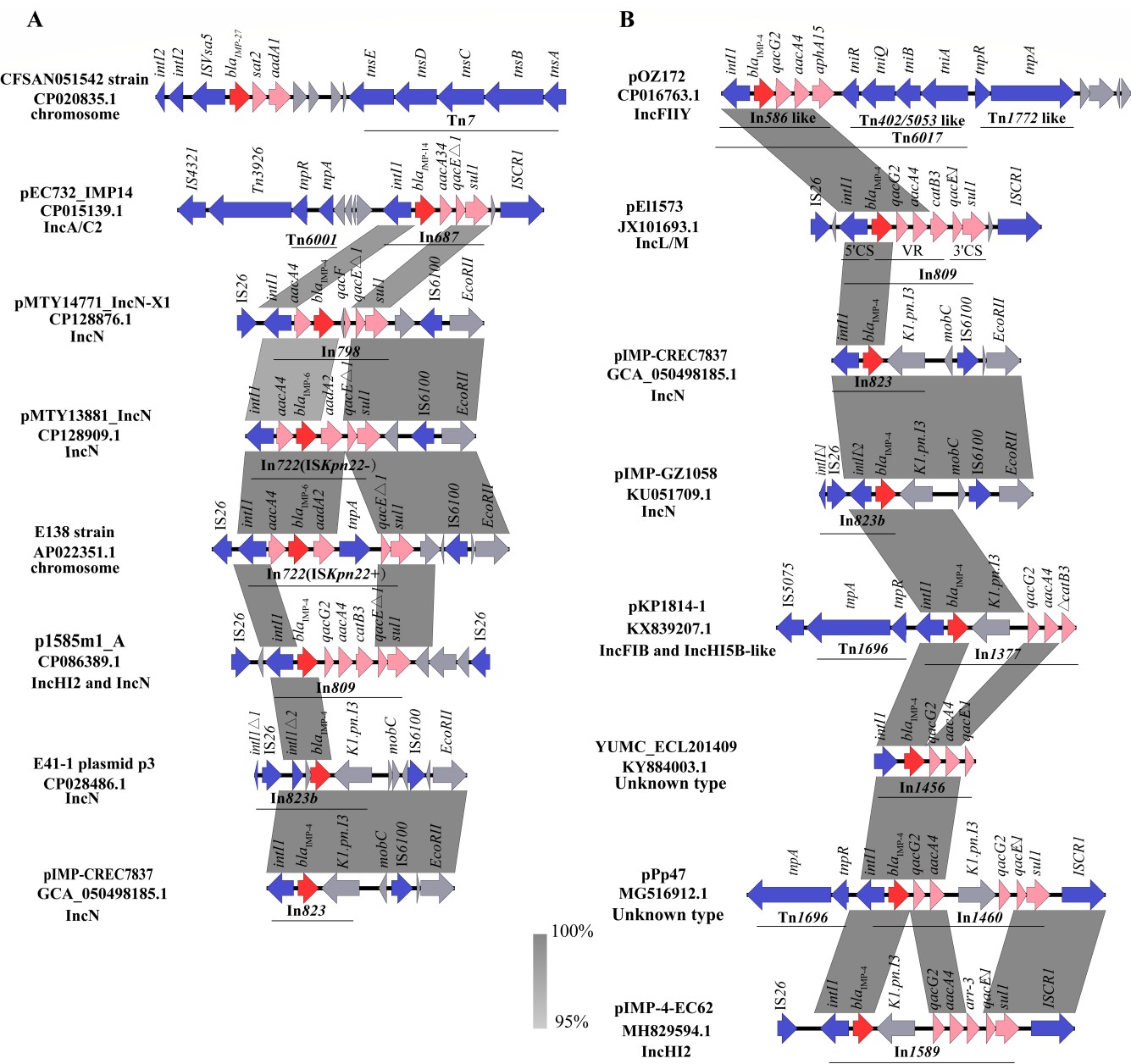

**FIG 4** Genetic characterization of the *bla*IMP in *Enterobacteriaceae*. (A) Schematic representation of different class 1 integrons carrying the *bla*IMP gene in *Escherichia coli*. (B) Schematic representation of different class 1 integrons carrying the *bla*IMP-4 gene in *Enterobacteriaceae*. Arrows indicate transcription directions of genes. The *bla*IMP genes are shown in red, and other resistant genes on class 1 integrons were labeled in pink; mobile elements were drawn in blue. The regions with highly similar sequences were marked in gray (nucleotide identity of ≥ 95%).

## Comparison of antimicrobial resistance genes, virulence factor genes, and plasmid replicons from different countries

Principal coordinate analysis (PCoA) was conducted to elucidate the distribution patterns of antimicrobial resistance genes (ARGs), virulence factor genes (VFs), and plasmids from the top 3 contributing countries of *bla*IMP-positive CRECs (Fig. 6A, C, and E). The results showed that the distribution of ARGs among isolates reported from China, Japan, and Australia clustered with relative independence and exhibited some diversity. In contrast, the distribution of VFs among isolates from the three countries showed similar patterns. Additionally, a comparable distribution of plasmids was observed between China and Japan. A Venn plot was used for co-existence analysis of ARGs, VFs, and plasmids from

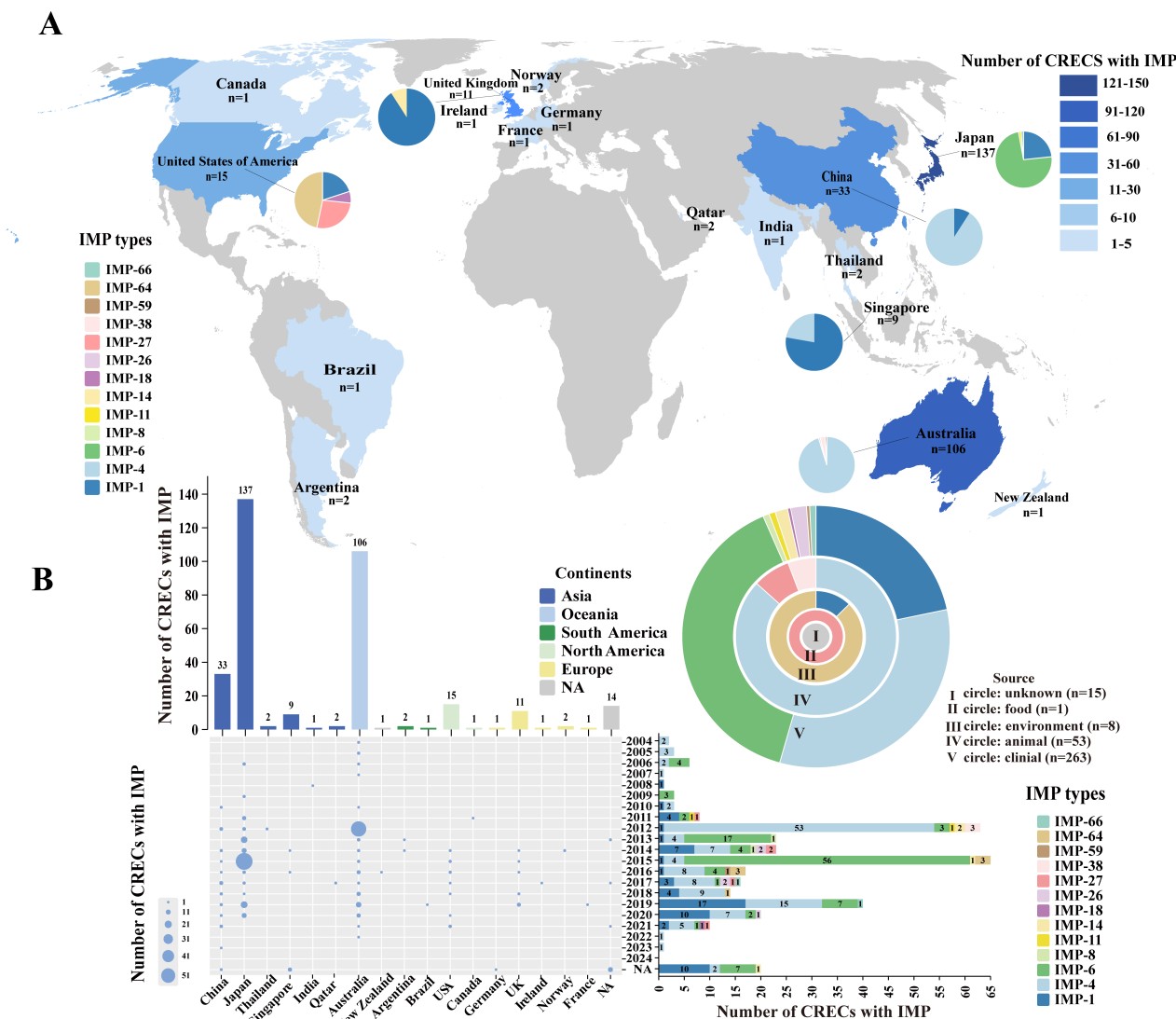

**FIG 5** Global surveillance of *bla*IMP-positive CRECs. (A) Global phylogeographic distribution of 340 *bla*IMP-positive CRECs, including the nine isolates in our study. The different colored blocks represent various quantity levels. Proportional distribution of enzymatic phenotypes across six countries (number of *bla*IMP-positive CRECs ≥ 5). Thirteen IMP enzymes are indicated by distinct colors. (B) Timeline of *bla*IMP-positive CREC outbreaks in 17 countries. The bubble size was proportional to the number of strains. The bar chart illustrates the quantities and proportions of the 13 enzymes. A circular plot displayed the distribution of 340 *bla*IMP-positive CRECs across various sources and proportions of the 13 enzymes. NA, unavailable relevant information.

the three countries (Fig. 6B, D, and F). The *bla*IMP-positive CRECs from Australia carried the highest and most unique numbers of ARGs, VFs, and plasmids, and the three countries showed relatively high overlap in ARGs, VFs, and plasmids, especially VFs.

## ST and phylogenetic analysis of *bla*IMP-positive CRECs

We conducted further analysis to evaluate the nine strains within a broader global context, as well as to understand the global epidemic STs, evolutionary characteristics, and other features of 340 *bla*IMP-positive CRECs. Among these 340 *bla*IMP-positive CREC strains, 82 STs have been identified, while remained 10 strains have untyped. We selected STs with an isolation count ≥5, as well as those widely reported in the literature with high resistance or virulence, for analysis and mapping. Over the past 20 years, prevalent ST strains have shown dynamic changes. From 2004 to 2010, the number of strains was small, with ST131 strains isolated during this period. ST131 was relatively prevalent during 2011–2015, with a significant increase in 2015. ST216 and ST58 strains increased

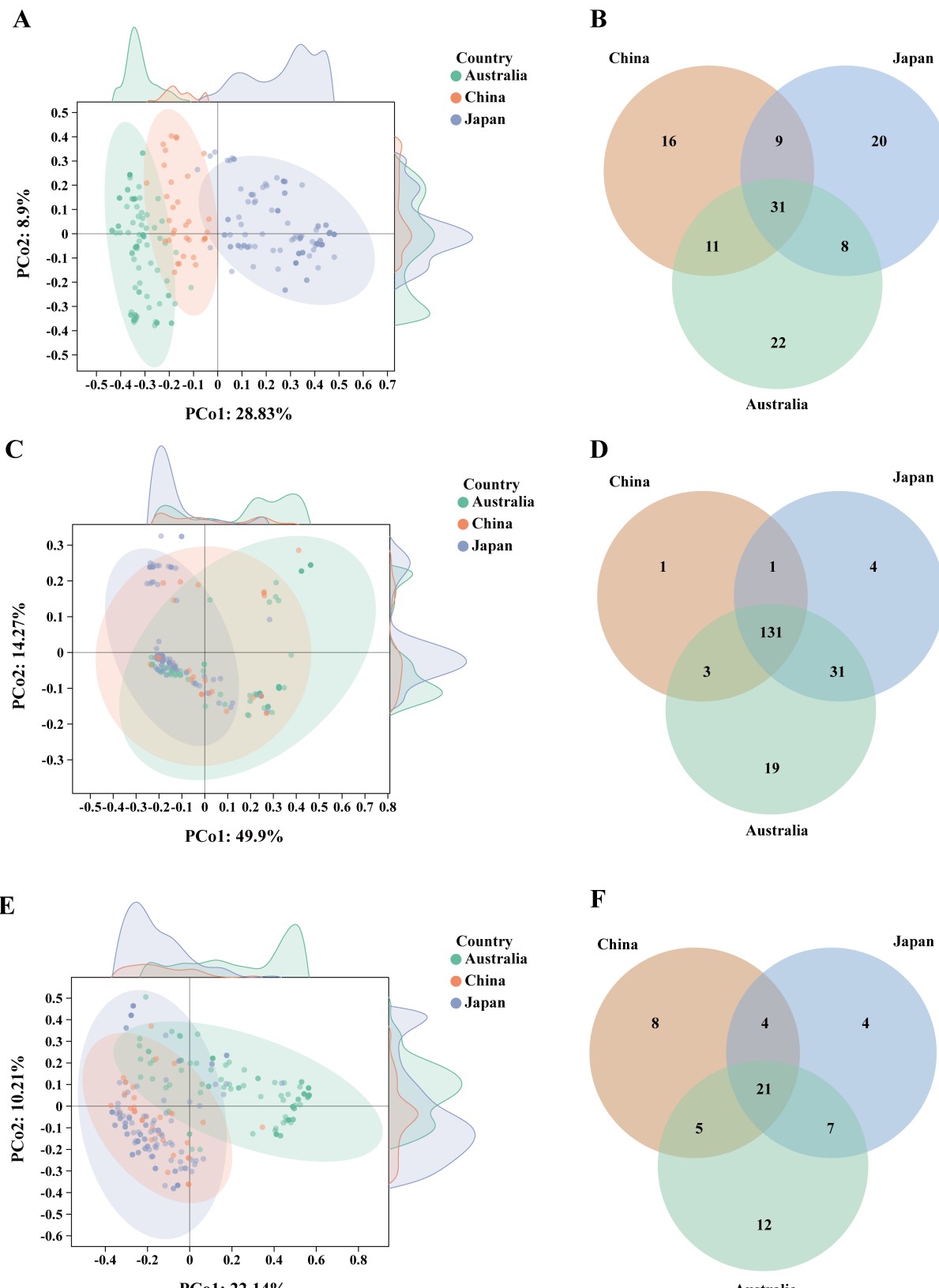

**FIG 6** Distribution characteristics of ARGs, VFs, and plasmid replicons of *bla*IMP-positive CRECs in China, Japan, and Australia (the data are listed in Tables S4 to S6). (A) PCoA of ARGs across the three countries. (B) Venn diagram of ARG types among the three countries. (C) PCoA of VFs across the three countries. (D) Venn diagram of VF types among the three countries. (E) PCoA of plasmids across the three countries. (F) Venn diagram of plasmid types among the three countries. Each dot in the figure represents a strain. Different colors represent different countries.

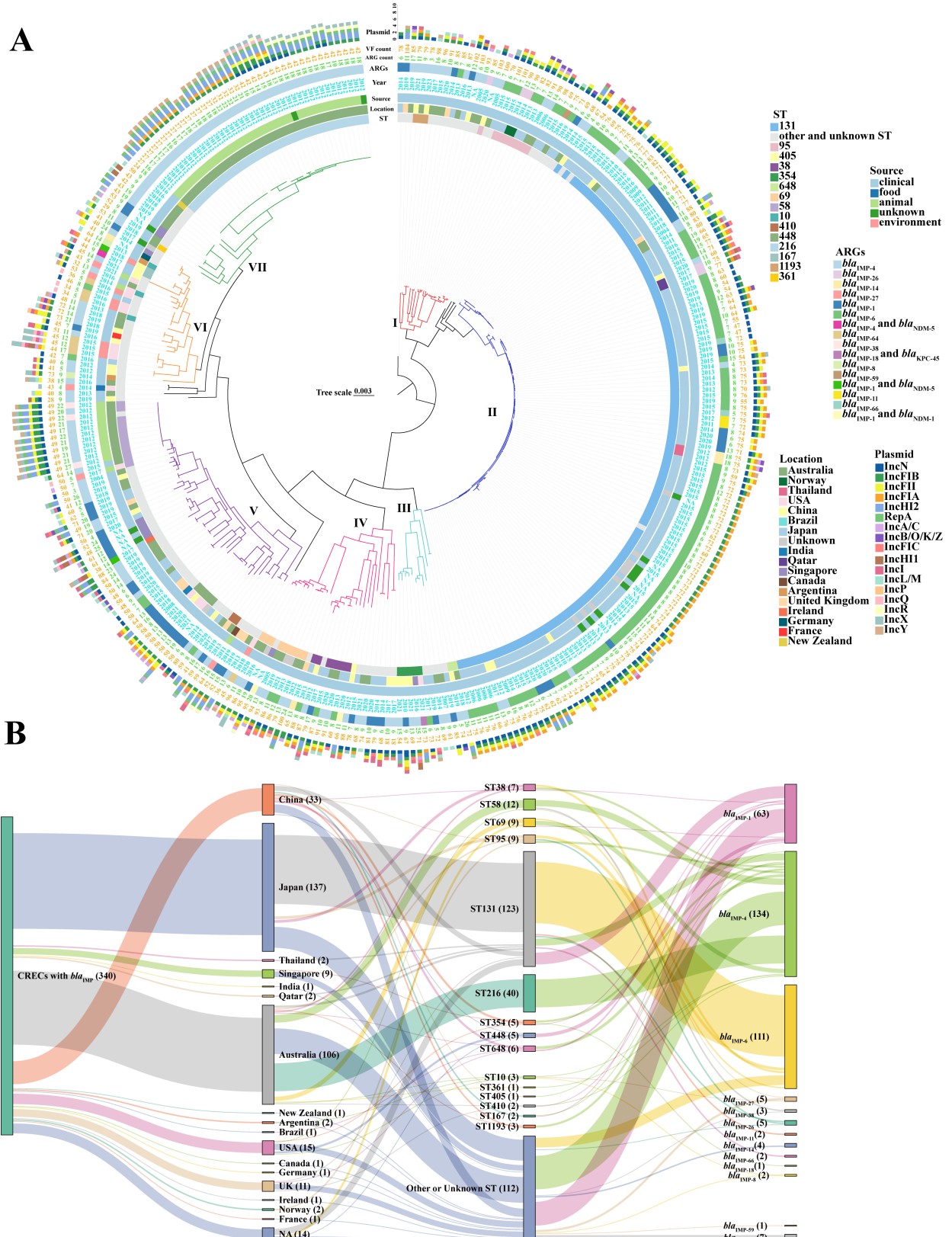

**FIG 7** (A) Phylogenetic tree of 340 $bla_{IMP}$-positive CRECs collected from this study and the NCBI database. Phylogroups (I–VII) represent the seven clones of $bla_{IMP}$-positive CRECs. (B) The Sankey diagram illustrated the relationships between STs and national distribution, as well as between STs and different $bla_{IMP}$ variants, among 340 $bla_{IMP}$-positive CRECs.

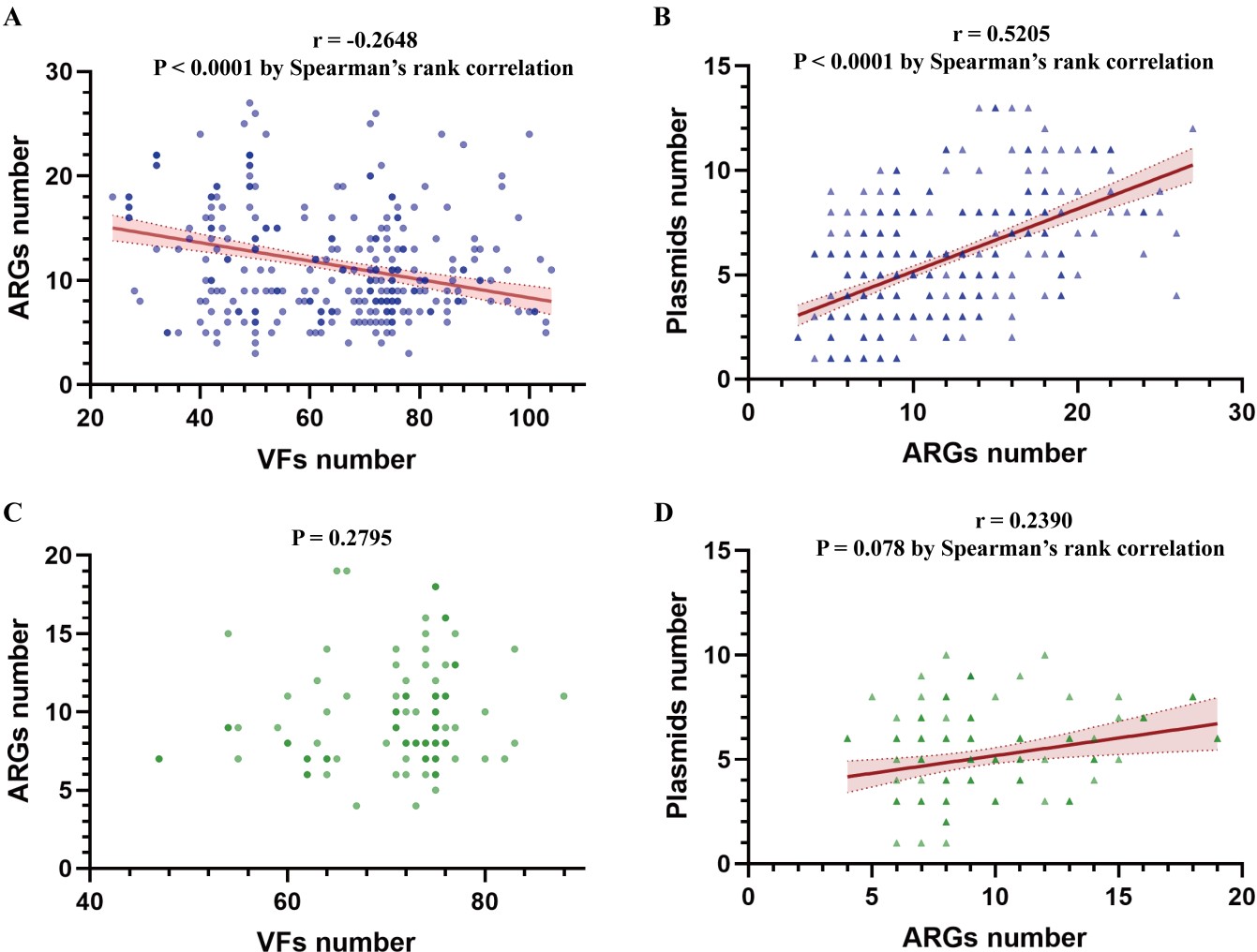

FIG 8 Correlation analysis of ARGs with VFs and plasmid replicons. (A) Spearman correlation analysis between ARGs and VFs. (B) Spearman correlation analysis between ARGs and plasmid replicons. (C) Spearman correlation analysis between ARGs and VFs in ST131. (D) Spearman correlation analysis between ARGs and plasmids in ST131.

in 2012. From 2016 to the present, the overall prevalence rate of ST131 has been lower than before; meanwhile, ST38, ST69, ST95, ST167, ST354, ST448, ST648, ST1193, and others have also been isolated occasionally, with no dominant ST emerging (Fig. S2).

The phylogenetic tree of 340 $bla_{IMP}$-positive CRECs can be roughly divided into seven evolutionary clades. The nine strains we collected were distributed across different evolutionary clades among the global $bla_{IMP}$-positive *E. coli*. The first *E. coli* carrying the $bla_{IMP-4}$ gene, isolated from a urine sample of an Australian patient in 2004, belonged to ST131 (Fig. 7A). ST131 strains clustered in the phylogenetic tree, belonging to clade II, accounting for 36.18% (123/340), and were the most common type. Most of these strains were mainly derived from Japan, China, and Australia, with scattered distribution in many other countries. ST131 strains were mainly derived from clinical patients, with the predominant $bla_{IMP-6}$, $bla_{IMP-1}$, and $bla_{IMP-4}$ (Fig. 7B). ST216 was dominant in Australia and only distributed in the country. ST216 clustered into clade VII in the phylogenetic tree, which only carried $bla_{IMP-4}$ and was isolated from animals in 2012. ST58 clustered into clade V in the phylogenetic tree, except one from the USA, others mainly isolated from wild animals of Australia in 2012. The variants in the ST58 strains were mainly $bla_{IMP-4}$ and $bla_{IMP-38}$, which differed by one amino acid substitution (Ser214Gly). It showed that local clone outbreak and $bla_{IMP}$ gene mutation occurred in ST58 strains.

ST216 and ST58 were clones prevalent in specific regions, relying on localized transmission routes and possessing the ability to adapt to the local environment (Fig. 7A). It can be observed that ST10, ST34, and ST58 have been isolated from both clinical patients and animals, while ST167 and ST218 have been isolated from both patients and the environment (Fig. 7A). This shows that the transmission of resistant bacteria is multi-source and complex, which increases the risk of their spread and poses a major challenge to public health.

## Correlation analysis of ARGs with VFs and plasmid replicons in $bla_{IMP}$-positive CRECs

Overall, we identified 149 different resistance genes, 206 different virulence genes, and 67 plasmid types among 340 $bla_{IMP}$-positive CRECs. There was a weak but statistically significant negative correlation between the number of resistance and virulence genes carried by each isolate ($r = -0.2648$; $P < 0.0001$ by Spearman's rank correlation) (Fig. 8A). Intriguingly, there was no correlation ($P = 0.2795$) between the number of resistance and virulence genes in ST131 (Fig. 8C). In addition, we found a positive correlation between the number of ARGs and plasmid replicons in each isolate ($r = 0.5205$, $P < 0.001$) and in ST131 ($r = 0.2390$, $P = 0.0078$), respectively (Fig. 8B and D). These results indicated that the plasmids played an important role in the transmission of ARGs. We also conducted a correlation analysis between STs and the number of resistance genes, virulence genes, and plasmids. The results showed that compared with the high-risk epidemic ST131, ST58 and ST216 carried a greater number of resistance genes and plasmids ($P < 0.05$), with no statistical differences observed in other STs; ST69 and ST95 carried more virulence genes ($P < 0.05$), while no statistical differences were found in ST38, ST354, and ST648 (Fig. S3). These STs in $bla_{IMP}$-positive *E. coli* require focused monitoring and management.

## DISCUSSION

In this study, we characterized phenotypic and genomic features of nine $bla_{IMP-4}$-positive CRECs from Zhejiang, China, and first reported the high-risk ST1193 clone carrying $bla_{IMP-4}$ gene. The nine multidrug-resistant strains showed resistant phenotypes, with polymyxin, tigecycline, and amikacin provided as last options for treatment. CREC5925 and CREC7387 were the same clone (ST1193), and CREC3005 and CREC3006 belonged to the same clone (ST973), while other strains showed high genetic variability with relatively scattered STs. However, the $bla_{IMP-4}$-carrying IncN plasmids among different clones were highly consistent, indicating that the IncN plasmid mediated the horizontal transfer of the $bla_{IMP-4}$ gene.

The IncN-type plasmid, known for its efficient conjugative properties, typically consists of four conjugative modules: *oriT*, a relaxase gene, the gene encoding the T4CP, and the gene cluster encoding the T4SS (20). In our collected strains, the IncN plasmid carried complete conjugation transfer system for self-transmission, with the pIMP-CREC7837 plasmid excluded due to IS26-mediated disruption of the *virB2* gene (21, 22). IS26 not only helps spread resistant genes but can also stop their spread by inserting randomly.

The broad-host-range IncN plasmids carry diverse resistance genes (e.g., to ESBLs, carbapenems, quinolones) and are key vectors in spreading carbapenemase genes (like $bla_{NDM}$, $bla_{KPC}$, and $bla_{IMP}$) among *Enterobacteriaceae* (e.g., *Escherichia* spp., *Klebsiella* spp., *Citrobacter* spp., and *Proteus* spp.) (23–27). Additionally, the efficient conjugation, stability, and low fitness cost of the IncN plasmids in our study underscore their role as effective $bla_{IMP}$ vectors. This is consistent with reports attributing $bla_{IMP-1/6}$ outbreaks in Japan to highly conjugative IncN-pST5 plasmids (28), and with findings that IncN plasmids (e.g., pIMP-1495) exhibit greater stability than F or HI2 types and impose minimal fitness cost (25, 27), collectively highlighting their critical function in $bla_{IMP}$ dissemination among *Enterobacteriaceae*.

The evolutionary trajectory of IncN plasmids may involve fusion with other types (e.g., IncX3, IncR), likely mediated by IS26, which enhances their stability, transferability, and resistance gene cargo (29–31). This potential is underscored by the diverse plasmid backgrounds (67 types identified) among global $bla_{IMP}$-CRECs, where conjugative plasmids like IncN, IncF, and IncHI2 are prevalent, facilitating $bla_{IMP}$ exchange and host range expansion.

Beyond plasmids, mobile genetic elements like class 1 integrons significantly facilitate $bla_{IMP}$ transmission. These integrons dynamically organize resistance gene cassettes, with expression levels influenced by cassette position relative to the promoter (19). In our sorted integrons, $bla_{IMP-4}$ was located in the first resistance gene cassette and was more readily expressed than other resistance genes. It was presumably associated with the widespread use of carbapenem antibiotics in recent years. Conversely, in In798 and In722, $bla_{IMP}$ followed aacA4 possibly reflecting aminoglycoside-driven selection. This cassette flexibility allows integrons to adapt rapidly to antibiotic pressures (32), making them valuable markers for $bla_{IMP}$ epidemiology.

Additionally, the integrons are frequently flanked by transposons or insertion sequences. This genetic context promoted $bla_{IMP}$ mobility, enabling its entry into vast plasmid clusters. For instance, In809 in Australian gulls promoted $bla_{IMP}$ dissemination via conjugative plasmids (5), while In823 in our strains formed a mobilizable unit with IS6100, serving as a resistance gene dissemination platform.

The $bla_{IMP-27}$ was chromosomally located within a Tn7-associated class 2 integron. In722 (ISKpn22−) was located on the IncN plasmid, while In722 (ISKpn22+) was located on the chromosome. Besides mediating the movement of integrons between plasmids and chromosomes (21), IS26 further drives integron evolution, as seen in pIMP-GZ1058 where it inactivated intI1, yet still facilitated resistance gene transfer. Similarly, p128379-IMP plasmid (MF344559.1) lacked the integrase intI1 but contained the IS26, which can also play a role in drug-resistant gene recombination, transfer, and expression (21). In summary, the integrons are recognized as the primary platform for the capture and expression of $bla_{IMP}$, while transposons and IS elements (particularly IS26) facilitate the structural evolution and mobilization of the entire resistance unit.

In global $bla_{IMP}$-positive CRECs, $bla_{IMP-6}$, $bla_{IMP-4}$, and $bla_{IMP-1}$ were predominant variants. Japan, China, and Australia were the top 3 contributors. Australia carried the highest number of ARGs, VFs, and plasmids. There were differences in the composition of ARGs among the three countries, indicating the regional evolution and dissemination processes of ARGs were relatively independent. The Spearman analysis showed plasmids play important roles in mediating the spread of resistance genes. In addition, we found that there is a certain degree of complementarity between the number of ARGs and VFs in the $bla_{IMP}$-positive CRECs, but ST131 is excluded. This suggests that bacteria need to maintain the balance of virulence and resistance to decrease a fitness cost (33, 34). But the high-risk epidemic ST131 has a flexible ability for adaptive evolution, with coevolution processes or independent acquisition pathways for its resistance genes and virulence genes (35–37). ST131 can possess both high drug resistance and virulence without excessively compromising its own survival. This trait may be attributed to the evolutionary advantages of this clone, which is also one of the reasons for it becoming a globally dominant clone.

MLST and phylogenetic analysis have revealed the genetic and evolutionary diversity of $bla_{IMP}$-positive CRECs. At present, 82 STs have been identified, and the phylogenetic tree can be roughly divided into seven evolutionary clades. Obviously, ST131 was the most common type. Due to increased use of carbapenems, the number of ST131 $bla_{IMP}$-positive CRECs began to increase around 2010, a trend that was consistent with the global spread of ST131 (38). In 2015, ST131 clone outbreaks may have occurred in Japan's clinical setting, contributing to the prevalence of $bla_{IMP-6}$. And the frequency of the IMP-6 phenotype was statistically significantly associated with usage of third-generation cephalosporins in Japan (39). ST131 $bla_{IMP}$-positive CRECs were not only highly prevalent in certain regions but also widely distributed globally. Additionally, we found

that the ST216 clone broke out in Australian silver gulls in 2012. A high incidence of IncHI2 plasmids carrying $bla_{IMP-4}$ was identified in this outbreak. The IncHI2 plasmids were similar to those from Australian human-derived clinical isolates, posing severe environmental pollution and public health risk (5).

Notably, we first reported two high-risk ST1193 clones carrying the $bla_{IMP-4}$ resistance gene. The sequence type is notable for its rapid rise to prominence across the globe over a short period of time and shares similar characteristics with ST131, mimicking the evolutionary trajectory of ST131 (38). An epidemiology study indicated that it was a significant cause of community-acquired urinary tract and bloodstream infections. ST1193 was the second most common AMR clone (behind ST131) in a geographically well-defined human population (38). ST1193 is universally resistant to fluoroquinolones due to quinolone resistance-determining region mutations and frequently associated with the ESBL gene ($bla_{CTX-M-15}$), which encodes resistance to most penicillins and cephalosporins (38). The emergence of ST1193 strains carrying carbapenem-resistant genes on mobilizable plasmids further poses challenges for clinical treatment. Global multidrug-resistant high-risk clones, including ST38, ST131, ST167, ST405, ST410, ST648, and ST1193, are all distributed in $bla_{IMP}$-positive CRECs (38). Additionally, the ST69, ST95, and ST131 have been confirmed closely associated with bloodstream infections (40). These pose potential challenges to antimicrobial stewardship and infection control. Appropriate monitoring measures should be implemented to prevent these high-risk clones, posing an increasingly serious threat to public health.

The limitation of this study is that, in NCBI, there are few records of strains isolated from 2022 to the present, making it impossible to reflect the prevalence of $bla_{IMP}$-positive CRECs in recent years. This may be related to the following factors: some countries or regions may not have performed genomic sequencing of strains due to economic reasons; in addition, the lag in uploading sequencing information may have led us to underestimate the prevalence of $bla_{IMP}$-positive CRECs.

## Conclusion

This study mainly describes the phenotypic and genomic characteristics of *E. coli* harboring $bla_{IMP-4}$-carrying IncN plasmids, which are responsible for the horizontal transfer of $bla_{IMP-4}$ gene among clones in the local region. It further clarifies the important role of other mobile genetic elements, such as class 1 integrons, IS*26*, and transposons, in the transmission of $bla_{IMP}$ resistance genes. Furthermore, our global genomic epidemiological study has elucidated the spatiotemporal distribution of $bla_{IMP}$-positive CRECs, providing a comprehensive understanding of their evolutionary trajectory and revealing high genetic diversity (82 STs), with ST131 predominant (36.18%). The regional evolution and dissemination processes of ARGs were relatively independent among China, Japan, and Australia. Correlation analysis indicated that plasmids play an important role in the transmission of ARGs. It is necessary to strengthen the monitoring of high-risk clones, such as ST131 and ST1193, reduce the resistance burden to carbapenem antibiotics, and prevent them from posing an increasingly serious threat to public health.

## MATERIALS AND METHODS

### Clinical isolation and resistance gene screening

From January 2017 to December 2024, a retrospective survey for 94 carbapenem-resistant *E. coli* isolates was identified from four tertiary hospitals in Zhejiang, China. Common carbapenemase genes ($bla_{KPC}$, $bla_{IMP}$, $bla_{VIM}$, $bla_{OXA-48}$, and $bla_{NDM}$) were amplified by polymerase chain reaction (PCR), and the $bla_{IMP}$-positive products were further whole-genome sequenced. Strain identification was performed using MALDI-TOF MS (bioMérieux, Marcy l'Etoile, France) (41).

## Antimicrobial susceptibility testing

The MICs of the nine $bla_{IMP-4}$-positive CRECs in this study were determined by the broth microdilution method, and the results using the breakpoints recommended by the 2025 Clinical and Laboratory Standards Institute guidelines for interpretation (42), except for tigecycline, which was interpreted according to the European Committee on Antimicrobial Susceptibility Testing guidelines (https://www.eucast.org/clinical_breakpoints). A total of 10 antibiotics were tested, including cefepime, ceftazidime, imipenem, meropenem, ertapenem, amikacin, ciprofloxacin, colistin, tigecycline, and ceftazidime-avibactam. *E. coli* ATCC 25922 was used as a quality control strain to ensure the accuracy of the experiment.

## Conjugative transfer experiments and conjugation frequencies

To explore the transferability of plasmids harboring $bla_{IMP-4}$, a conjugative transfer assay was performed, as previously described (43). First, monoclonal colonies of donor and recipient *E. coli* J53 bacteria were collected into 2 mL of Luria-Bertani (LB) broth (Sangon Biotech, Shanghai, China), reached the logarithmic growth phase, mixed at a ratio of 1:1, and cultured on MH plates with a filter membrane overnight at 37°C. Transconjugants were screened on MH agar containing sodium azide (200 µg/mL) and ampicillin (100 µg/mL). Finally, the transconjugants were confirmed by PCR, MALDI-TOF MS, and MIC results.

To calculate conjugation frequency, we immersed and mixed the above overnight-cultured filter membranes in 0.9% NaCl solution. Then, we diluted the mixture to the appropriate concentration gradient and spread it on two types of MH agar plates: one containing both ampicillin (100 µg/mL) and sodium azide (200 µg/mL), and another containing only ampicillin (100 µg/mL). The conjugation frequency was calculated by dividing the number of transconjugants (CFU/mL) by the number of donor strains (CFU/mL), as previously described (44).

## Stability of the plasmid

To determine the stability of plasmids in transconjugants, a passaging experiment was performed in a non-antibiotic environment (45). Three monoclonal transconjugants were inoculated in 2 mL of LB broth without antibiotics at 37°C with constant shaking. Then, 2 µL of overnight culture was serially passed into 2 mL of fresh LB broth every day for 10 days. Overnight cultures were diluted to appropriate concentration gradients every 5 days and then spread on non-selective MH plates. Fifty monoclones were selected from each plate for PCR to verify whether the plasmid was lost.

## Growth rate determination

We determined the growth rate of transconjugants to describe their fitness cost. Four monoclonal transconjugants were grown overnight. The overnight cultures, with a 1:100 dilution ratio in LB broth, were added to a flat-bottom 100-well plate; each overnight culture was repeated three times. The plates were incubated at 37°C with shaking at 200 rpm. The optical density of each culture at 600 nm was measured every 5 min for 20 h by Bioscreen C analyzer (Oy Growth Curves Ab. Ltd., Finland). The relative growth rate was estimated based on $OD_{600}$ curves using an R script, as previously described (46). Both the growth curve and the area under the curve were derived from these $OD_{600}$ measurements. Then, they were analyzed and visualized by GraphPad Prism version 9.

## Whole-genome sequencing and genomic analysis

WGS was conducted using the Illumina HiSeq and Nanopore MinION platforms at Zhejiang Tianke (Hangzhou, China). Complete genome sequences were assembled using the hybrid assembly tool Unicycler 0.4.8 (47) and annotated using RAST (https://rast.nmpdr.org/). ResFinder v.4.6.0 and PlasmidFinder v.2.0.1 from the Center of Genomic Epidemiology (https://www.genomicepidemiology.org/services/) were used to identify

antibiotic resistance genes and plasmid types, respectively. The Orit Finder website (https://bioinfo-mml.sjtu.edu.cn/oriTfinder/) was used to forecast the self-transfer or mobilizing capability of bacterial mobile genetic elements (20). BLAST from NCBI (https://blast.ncbi.nlm.nih.gov/Blast.cgi/) was used to find similar sequences of plasmids and genes. Sequence comparisons were performed using BLASTn v.2.4.0 and visualized using Easyfig v.2.2.3 and Proksee (https://proksee.ca/). Heatmaps were utilized for the visualization of SNPs using ChiPlot (https://www.chiplot.online/).

## Global phylogeographic, MLST, and phylogenetic analysis of $bla_{IMP}$ in *E. coli*

In a total of 486,101 *E. coli* strains from NCBI, we screened 340 non-redundant, fully assembled whole-genome sequences of $bla_{IMP}$-positive CRECs, including nine strains in our study (https://www.ncbi.nlm.nih.gov/pathogens/isolates, accessed on 9 July 2025). Species were delineated based on an average nucleotide identity (ANI) >95% using FastANI (https://github.com/ParBLiSS/FastANI). A total of 340 CRECs harboring antibiotic resistance genes, virulence genes, and the Inc-type plasmid of the strain were screened and identified using ABRicate (https://github.com/tseemann/abricate). Multilocus sequence typing was performed using the MLST tool (https://github.com/tseemann/mlst). The phylogenetic tree was constructed using Roary and FastTree and further visualized using ChiPlot (https://www.chiplot.online/). Adobe Illustrator v.27.9.1 was used to map the global geographic distribution.

## Statistical analysis

We performed principal coordinate analysis based on the Jaccard algorithm and constructed confidence circles at a 95% confidence level to evaluate the composition of ARGs, VFs, and plasmids in $bla_{IMP}$-positive CRECs from China, Japan, and Australia. (https://www.chiplot.online/) (16, 17). The Venn plot was used on the online website (https://www.bioinformatics.com.cn/static/others/jvenn/example.html) for co-existence analysis of three countries. GraphPad Prism version 9 was utilized for Spearman analysis and visualization of ARGs, VFs, and plasmid replicons. An adjusted *P*-value <0.05 was considered significant. Absolute *r*-values indicated correlation strength: 0.5–1.0 (high), 0.3–0.5 (moderate), 0.1–0.3 (low), and <0.1 (no correlation) (48).

## ACKNOWLEDGMENTS

This study was supported by the National Natural Science Foundation of China (82472323 and 82172306), the Zhejiang Provincial Natural Science Foundation Outstanding Youth Program (LR25H200001), the National Health Commission Scientific Research Fund-Zhejiang Provincial Major Health Science and Technology Plan Project (WKJ-ZJ-2414), and the Research and Development Program of Zhejiang Province (2023C03068).

Y.H.: Writing-original draft, methodology, data curation, and visualization. C.W.: Methodology and investigation. J.Y.: Conceptualization and data curation. J.G.: Software and visualization. J.S., X.W., L.Z., X.T., and H.X.: Investigation and data curation. Y.Y.: Resources. X.L.: Methodology and funding acquisition. X.H.: Writing-review and editing. All authors have read and agreed to the published version of the manuscript.

## AUTHOR AFFILIATIONS

[1]Wenzhou Medical University, Wenzhou, Zhejiang, China
[2]Centre of Laboratory Medicine, Zhejiang Provincial People's Hospital, Affiliated People's Hospital, Hangzhou Medical College, Hangzhou, Zhejiang, China
[3]Department of Clinical Laboratory, Zhejiang Cancer Hospital, Hangzhou Institute of Medicine (HIM), Chinese Academy of Sciences, Hangzhou, Zhejiang, China

## AUTHOR ORCIDs

Yunsong Yu  http://orcid.org/0000-0003-2903-918X
Xi Li  http://orcid.org/0000-0002-8838-7814
Xinhong Han  http://orcid.org/0009-0004-4125-9075

## AUTHOR CONTRIBUTIONS

Yueyue Hu, Data curation, Methodology, Visualization, Writing – original draft | Chengjin Wu, Data curation, Investigation, Methodology, Visualization, Writing – original draft | Jiayao Yao, Conceptualization, Data curation, Investigation, Methodology, Software | Jingyi Guo, Conceptualization, Data curation, Methodology, Software | Jie Sheng, Data curation, Investigation, Software | Xinru Wang, Data curation, Investigation | Longjie Zhou, Data curation, Investigation | Xinyan Tang, Investigation | Haotian Xu, Investigation | Yunsong Yu, Investigation, Resources | Xi Li, Conceptualization, Funding acquisition, Methodology, Resources, Supervision | Xinhong Han, Conceptualization, Funding acquisition, Methodology, Supervision, Writing – review and editing

## DATA AVAILABILITY

All data generated or analyzed in this study are available in the main article and the appendix. Complete or draft genome assemblies were deposited in the NCBI database with accession numbers GCA_050498225.1, GCA_050498105.1, GCA_039580285.1, GCA_050498165.1, GCA_050498145.1, GCA_050498205.1, GCA_050498185.1, GCA_050498245.1, GCA_036916335.1.

## ETHICS APPROVAL

This study was conducted in accordance with the Declaration of Helsinki and had been reviewed and approved by the Research Ethics Committee of Zhejiang Provincial People's Hospital (Approval no.: QT2025090).

## ADDITIONAL FILES

The following material is available online.

### Supplemental Material

**Figure S1 (Spectrum03244-25-s0001.tif).** Fig. S1: Cluster analysis of nine $bla_{IMP}$-positive CRECs based on single nucleotide polymorphisms (SNP).
**Figure S2 (Spectrum03244-25-s0002.tif).** Fig. S2: Bar graph of the number of some STs over time.
**Figure S3 (Spectrum03244-25-s0003.tif).** Fig. S3: A correlation analysis between ST types and the number of resistance genes, virulence genes.
**Figure S1 to S3 legends (Spectrum03244-25-s0004.docx).** Fig. S1: Cluster analysis of nine $bla_{IMP}$-positive CRECs based on single nucleotide polymorphisms (SNP). Fig. S2: Bar graph of the number of some STs over time. Fig. S3: A correlation analysis between ST types and the number of resistance genes, virulence genes.
**Tables S1 to S6 (Spectrum03244-25-s0005.xlsx).** Information of our collected strains and 340 IMP-producing CRECs

### Open Peer Review

**PEER REVIEW HISTORY (review-history.pdf).** An accounting of the reviewer comments and feedback.

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
