## [Reviewer comments · Microbiology Spectrum]

Microbiology Spectrum

Global epidemiological and genetic characteristics of carbapenem-resistant *Escherichia coli* carrying *bla_{IMP}*

Yueyue Hu, Chengjin Wu, Jiayao Yao, Jingyi Guo, Jie Sheng, Xinru Wang, Longjie Zhou, Xinyan Tang, Haotian Xu, Yunsong Yu, Xi Li, and Xinhong Han

Corresponding Author(s): Xinhong Han, Zhejiang Cancer Hospital, Hangzhou Institute of Medicine (HIM), Chinese Academy of Sciences

Review Timeline:

Submission Date:	October 9, 2025
Editorial Decision:	October 26, 2025
Revision Received:	October 31, 2025
Accepted:	November 12, 2025

Editor: Haifang Zhang

Reviewer(s): Disclosure of reviewer identity is with reference to reviewer comments included in decision letter(s). The following individuals involved in review of your submission have agreed to reveal their identity: Fuping HU (Reviewer #2)

Transaction Report:

DOI: <https://doi.org/10.1128/spectrum.03244-25>

Re: Spectrum03244-25 (**Global epidemiological and genetic characteristics of carbapenem-resistant *Escherichia coli* carrying bla_{IMP}**)

Dear Dr. Xinhong Han:

Thank you for the privilege of reviewing your work. Below you will find my comments, instructions from the Spectrum editorial office, and the reviewer comments.

Revision Guidelines

Sincerely,
Haifang Zhang
Editor
Microbiology Spectrum

Reviewer #1 (Comments for the Author):

This study provides a comprehensive genomic and epidemiological analysis of bla_{IMP}-positive carbapenem-resistant *E. coli* (CREC), integrating local surveillance data from Zhejiang, China, with global genomic datasets. The work addresses a significant gap in understanding the dissemination mechanisms and evolutionary dynamics of bla_{IMP}-positive CREC. The combination of phenotypic assays, plasmid characterization, and global phylogenetic analysis strengthens the study's impact. The findings on

high-risk ST1193 clones, plasmid stability, and the role of mobile genetic elements are particularly valuable. Below, I provide several specific comments to further enhance this manuscript.

1, The manuscript effectively establishes the IncN plasmid as the dominant vector for blaIMP- dissemination. To fully contextualize this critical finding, the Discussion should be expanded to address the broader context of the IncN replicon. Specifically, the authors should briefly review IncN's known global role in spreading other crucial antimicrobial resistance genes across Enterobacteriaceae. Furthermore, a review of existing literature regarding the reported adaptive costs and stability of IncN plasmids across different clonal backgrounds is needed.

2, Discussion Paragraphs 2 and 3 are overly long and should be streamlined. The authors are strongly encouraged to divide these long paragraphs into smaller, focused units, each presenting a single major point.

3, While blaIMP-positive ST1193 is first reported as a high-risk clone, more context on its global clinical relevance (e.g., association with community-acquired infections) would strengthen the discussion. Briefly cite key studies on ST1193's epidemiology to support its public health threat.

4, The correlation analysis is a powerful finding. While the correlation analysis for all isolates showed a weak negative correlation between ARGs and VFs, ST131 showed no correlation. Should it be directly linked to the clone's high evolutionary success or sampling bias?

Reviewer #2 (Comments for the Author):

The manuscript presents a highly valuable study that comprehensively investigates the phenotypic and genomic characteristics of blaIMP-positive Carbapenem-Resistant E. coli (CREC). The integration of local surveillance data from China with global genomic analysis of 340 isolates provides a comprehensive framework for understanding the current epidemiology and transmission dynamics of blaIMP. The manuscript described the high-risk ST1193 clone carrying blaIMP-4 for the first time. Besides, the detailed characterization of the IncN plasmids offers critical insights for public health surveillance and future antibiotic stewardship programs. The paper is well-structured and suitable for publication after minor revisions.

1, The third paragraph of discussion (Line 344-377) about mobile genetic elements (MGEs) is structurally fragmented. Please restructure this paragraph to clearly discuss the concept of MGE collaboration in blaIMP dissemination. Integrons should be established as the primary platform for blaIMP capture and expression. Then, transposons and IS elements (particularly IS26) drive the structural evolution and movement of the entire resistance unit. This will make the argument significantly more concise.

2, Please check and correct a minor inconsistency regarding the clonal grouping in the text. The Results section (Line 128) states that CREC5925 and CREC7387 were the same clone (ST1193). However, the Discussion section (Line 317) states that CREC5925 and CREC7837 belonged to the same clone. Please verify which strain ID is correct.

3, Please perform a final check for minor typographical errors throughout the manuscript, such as the misspelling of "reaistance" (Line 386) and the lack of a space in "phylogeneticanalysis" (Line 391).

Dear editor and reviewers:

Thank you for taking our manuscript "**Global epidemiological and genetic characteristics of carbapenem-resistant *Escherichia coli* carrying *bla*_{IMP}**" (Spectrum03244-25) into consideration. We appreciate your comments and suggestions which are valuable for improving our paper and helpful in promoting the importance of our work. According to the comments and suggestions, a word-by-word revision has been made on the manuscript and highlighted in the revised version. We make a point-by-point response to the comments in this letter and hope that our revised manuscript meets the requirements for publication in "*Microbiology Spectrum*".

Reviewer #1 (Comments for the Author):

This study provides a comprehensive genomic and epidemiological analysis of blaIMP-positive carbapenem-resistant E. coli (CREC), integrating local surveillance data from Zhejiang, China, with global genomic datasets. The work addresses a significant gap in understanding the dissemination mechanisms and evolutionary dynamics of blaIMP-positive CREC. The combination of phenotypic assays, plasmid characterization, and global phylogenetic analysis strengthens the study's impact. The findings on high-risk ST1193 clones, plasmid stability, and the role of mobile genetic elements are particularly valuable. Below, I provide several specific comments to further enhance this manuscript.

1, The manuscript effectively establishes the IncN plasmid as the dominant vector for blaIMP- dissemination. To fully contextualize this critical finding, the Discussion should be expanded to address the broader context of the IncN replicon. Specifically, the authors should briefly review IncN's known global role in spreading other crucial antimicrobial resistance genes across Enterobacteriaceae. Furthermore, a review of existing literature regarding the reported adaptive costs and stability of IncN plasmids across different clonal backgrounds is needed.

Response: Thank you for your insightful comments on our manuscript. By integrating relevant literature with the data from this study, we conducted a deeper discussion and supplemented insights into IncN plasmids—covering **their complete conjugative transfer systems, broad host ranges, and carriage of diverse resistance genes**, as well as the **conjugative capacity, stability, and fitness cost of IncN plasmids** across different clonal backgrounds—thus clarifying the mechanism underlying IncN plasmids' role in resistance gene transmission.

The revised content is located in Lines 319-341.

2, Discussion Paragraphs 2 and 3 are overly long and should be streamlined. The authors are strongly encouraged to divide these long paragraphs into smaller, focused units, each presenting a single major point.

Response: Thank you for your insightful comments on our manuscript. We reorganized the logical structure of paragraphs 2 and 3 in the original article, streamlined their content, and split them into smaller paragraphs—with each paragraph elaborating on one key point.

Lines319-325: Describes the standard conjugation system of IncN plasmids and how an IS26 insertion in one plasmid (pIMP-CREC7837) disrupted this function.

Lines326-335: Summarizes the role of broad-host-range IncN plasmids as key vectors for spreading carbapenemase genes, supported by their efficient conjugation, stability, and low fitness cost observed in this and other studies.

Lines336-341: Proposes that IncN plasmids may evolve through IS26-mediated fusion with other plasmid types, a potential highlighted by the diverse plasmid backgrounds found in global *bla*_{IMP}-CREC isolates.

Lines342-351: *Bla*_{IMP} is frequently present in class 1 integrons. The gene cassette array of class 1 integrons was dynamic and diverse under antimicrobial pressure.

Lines352-367: Integrons are recognized as the primary platform for the capture and expression of *bla*_{IMP}, while transposons and IS elements facilitate its mobilization.

3, While *bla*_{IMP}-positive ST1193 is first reported as a high-risk clone, more context

on its global clinical relevance (e.g., association with community-acquired infections) would strengthen the discussion. Briefly cite key studies on ST1193's epidemiology to support its public health threat.

Response: Thank you for your valuable suggestion. We cited studies on the epidemiology and drug resistance of ST1193 (Lines 399-409), demonstrating its risk to public health.

The context is supplemented in Lines 399-409 as: The sequence type is notable for its rapid rise to prominence across the globe over a short period of time, and shares similar characteristics with ST131, mimicking the evolutionary trajectory of ST131 [38]. An epidemiology study indicated that it was a significant cause of community-acquired urinary tract and bloodstream infections. And ST1193 was the second most common AMR clone (behind ST131) in a geographically well-defined human population [38]. ST1193 is universally resistant to fluoroquinolones due to quinolone resistance-determining region (QRDR) mutations and frequently associated with the ESBL gene (*bla*_{CTX-M-15}), which encodes resistance to most penicillins and cephalosporins [38]. The emergence of ST1193 strains carrying carbapenem-resistant genes on mobilizable plasmids further poses challenges for clinical treatment.

4, The correlation analysis is a powerful finding. While the correlation analysis for all isolates showed a weak negative correlation between ARGs and VFs, ST131 showed no correlation. Should it be directly linked to the clone's high evolutionary success or sampling bias?

Response: Thanks for your insightful questions. According to existing literature, the resistance genes and virulence genes of ST131 strains show no correlation—a phenomenon attributed to the strain's complex evolution. The evolution of these two gene types may involve a coevolutionary mechanism or proceed via independent pathways, and the detailed mechanisms need to be further studied. We base this conclusion on the following evidence from our manuscript:

Dominant Sample Size: ST131 is not a minor or rare clone in our dataset. It is the most predominant sequence type, accounting for 36.18% (123/340) of all isolates.

This large and geographically diverse collection, with isolates from Japan, China, Australia, and other countries. A unique pattern observed in such a large, dominant clone is less likely to be a simple sampling error.

Biological Plausibility: This finding is highly consistent with the known biology of ST131 as a globally dominant, high-risk clone. In the general *E. coli* population, the weak negative correlation we found suggests a fitness cost, that makes it difficult to maintain high numbers of both ARGs and VFs.

Adaptive Flexibility: As we state in our discussion, ST131 appears to possess a "flexible ability for adaptive evolution". This allows it to acquire resistance genes and virulence genes through independent pathways. Its ability to accumulate both high resistance and high virulence without a major fitness cost is likely a primary reason for its global success.

We agree with the reviewer that sampling bias is an important consideration in any study using public databases like NCBI, which are not the result of a single, systematic surveillance program.

The discussion was added in Lines 377-382.

Reviewer #2 (Comments for the Author):

The manuscript presents a highly valuable study that comprehensively investigates the phenotypic and genomic characteristics of blaIMP-positive Carbapenem-Resistant *E. coli* (CREC). The integration of local surveillance data from China with global genomic analysis of 340 isolates provides a comprehensive framework for understanding the current epidemiology and transmission dynamics of blaIMP. The manuscript described the high-risk ST1193 clone carrying blaIMP-4 for the first time. Besides, the detailed characterization of the IncN plasmids offers critical insights for public health surveillance and future antibiotic stewardship programs. The paper is well-structured and suitable for publication after minor revisions.

1, The third paragraph of discussion (Line 344-377) about mobile genetic elements (MGEs) is structurally fragmented. Please restructure this paragraph to clearly discuss

the concept of MGE collaboration in blaIMP dissemination. Integrons should be established as the primary platform for blaIMP capture and expression. Then, transposons and IS elements (particularly IS26) drive the structural evolution and movement of the entire resistance unit. This will make the argument significantly more concise.

Response: Thank you for your valuable suggestion. We reorganized the logical structure of paragraphs 3 in the original article and explained from the following two aspects.

Lines342-351: *Bla*_{IMP} is frequently present in class 1 integrons. The gene cassette array of class 1 integrons was dynamic and diverse under antimicrobial pressure.

Lines352-367: Integrons are recognized as the primary platform for the capture and expression of *bla*_{IMP}, while transposons and IS elements facilitate its mobilization.

2, Please check and correct a minor inconsistency regarding the clonal grouping in the text. The Results section (Line 128) states that CREC5925 and CREC7387 were the same clone (ST1193). However, the Discussion section (Line 317) states that CREC5925 and CREC7837 belonged to the same clone. Please verify which strain ID is correct.

Response: Thank you for pointing out the errors. We have made a more accurate description.

Lines 314-316: CREC5925 and CREC7387 were the same clone (ST1193), and CREC3005 and CREC3006 belonged to same clone (ST973), while other strains showed high genetic variability with relatively scattered ST types.

3, Please perform a final check for minor typographical errors throughout the manuscript, such as the misspelling of "reaistance" (Line 386) and the lack of a space in "phylogeneticanalysis" (Line 391).

Response: Thank you for pointing out the spelling errors. We have corrected them, please see **Line 376 and Line 383**.

Finally, we have carefully reviewed the manuscript and polished the language. We appreciate for reviewers' warm work earnestly, and hope that the correction will meet with approval. Once again, thank you very much for your comments and suggestions.

Re: Spectrum03244-25R1 (**Global epidemiological and genetic characteristics of carbapenem-resistant *Escherichia coli* carrying *bla*_{IMP}**)

Dear Dr. Xinhong Han:

Your manuscript has been accepted, and I am forwarding it to the ASM production staff for publication. Your paper will first be checked to make sure all elements meet the technical requirements. ASM staff will contact you if anything needs to be revised before copyediting and production can begin. Otherwise, you will be notified when your proofs are ready to be viewed.

Sincerely,
Haifang Zhang
Editor
Microbiology Spectrum